# Development of an exon skipping therapy for X-linked Alport syndrome with truncating variants in *COL4A5*

Tomohiko Yamamura[1], Tomoko Horinouchi[1], Tomomi Adachi[2], Maki Terakawa[2], Yutaka Takaoka[3], Kohei Omachi[4], Minoru Takasato[5], Kiyosumi Takaishi[2], Takao Shoji[6], Yoshiyuki Onishi[6], Yoshito Kanazawa[6], Makoto Koizumi[6], Yasuko Tomono[7], Aki Sugano[3], Akemi Shono[1], Shogo Minamikawa[1], China Nagano[1], Nana Sakakibara[1], Shinya Ishiko[1], Yuya Aoto[1], Misato Kamura[4], Yutaka Harita[8], Kenichiro Miura[9], Shoichiro Kanda[8], Naoya Morisada[1], Rini Rossanti[1], Ming Juan Ye[1], Yoshimi Nozu[1], Masafumi Matsuo[10], Hirofumi Kai[4], Kazumoto Iijima[1] & Kandai Nozu[1]✉

Currently, there are no treatments for Alport syndrome, which is the second most commonly inherited kidney disease. Here we report the development of an exon-skipping therapy using an antisense-oligonucleotide (ASO) for severe male X-linked Alport syndrome (XLAS). We targeted truncating variants in exon 21 of the *COL4A5* gene and conducted a type IV collagen α3/α4/α5 chain triple helix formation assay, and in vitro and in vivo treatment efficacy evaluation. We show that exon skipping enabled trimer formation, leading to remarkable clinical and pathological improvements including expression of the α5 chain on glomerular and the tubular basement membrane. In addition, the survival period was clearly prolonged in the ASO treated mice group. This data suggests that exon skipping may represent a promising therapeutic approach for treating severe male XLAS cases.

---

[1] Department of Pediatrics, Kobe University Graduate School of Medicine, Kobe, Japan. [2] Rare Disease Laboratories, Daiichi Sankyo Co., Ltd., Shinagawa, Tokyo, Japan. [3] Division of Medical Informatics and Bioinformatics, Kobe University Hospital, Kobe, Japan. [4] Department of Molecular Medicine, Graduate School of Pharmaceutical Sciences, Kumamoto University, Kumamoto, Japan. [5] RIKEN Center for Developmental Biology, Kobe, Japan. [6] Modality Research Laboratories, Daiichi Sankyo Co., Ltd., Shinagawa, Tokyo, Japan. [7] Division of Molecular Cell Biology, Shigei Medical Research Institute, Okayama, Japan. [8] Department of Pediatrics, Graduate School of Medicine, The University of Tokyo, Tokyo, Japan. [9] Department of Pediatric Nephrology, Tokyo Women's Medical University, Tokyo, Japan. [10] Department of Physical Therapy, Faculty of Rehabilitation, Kobe Gakuin University, Kobe, Japan. ✉email: nozu@med.kobe-u.ac.jp

Alport syndrome (AS) is a progressive inherited nephritis accompanied by sensorineural loss of hearing and ocular abnormalities. AS develops because of pathogenic variants in one of the three type IV collagen encoding genes, *COL4A3*, *COL4A4*, and *COL4A5*[1,2]. These three genes encode type IV collagen α3–α5 chains (α3(IV)–α5(IV)), respectively, and those chains assemble to form triple helix structures (α345(IV)) that combine in the glomerular basement membrane (GBM). Formation of these triple helix structures is disrupted when an abnormality occurs in one of the three α chains, and this causes AS[3]. AS is divided into three groups according to the inheritance modes: X-linked Alport syndrome (XLAS), autosomal recessive AS, and autosomal dominant AS. XLAS is caused by pathogenic variants in the *COL4A5* gene and according to the character of the X-linked disease, male XLAS cases show much severer phenotypes and develop end-stage renal disease (ESRD) during their 20s and 30s[4–6]. There is currently no radical therapy for this disease and treatment with nephron-protective drugs only delays progression to ESRD. For example, angiotensin-converting enzyme inhibitors can remarkably delay the development of ESRD by ~10 years[7]. However, male XLAS cases show a strong genotype–phenotype correlation and patients possessing truncating mutations still show severe phenotypes and develop ESRD around the age of 20[4–6]. Therefore, the development of new treatments for cases with truncating variants is needed urgently.

Antisense oligonucleotides (ASOs) have been successfully used to treat various inherited diseases, including Nusinersen for spinal muscular atrophy (SMA), Eteplirsen for Duchenne muscular dystrophy (DMD), Mipomersen for hypercholesterolemia, and Inotersen for amyloidosis. We recently published data showing that *COL4A5* gene splice site mutations with an in-frame deletion

at the transcript level showed good renal prognosis and ESRD developed after 9 years when compared with the out-of-frame deletion group[8]. Furthermore, we also reported a 47-year-old male with a deletion of 105 bp at the transcript level because of a splice site mutation, who had still not developed ESRD[9]. These data prompted us to develop exon-skipping therapy for male XLAS cases with truncation mutations. The *COL4A5* gene (NM_000495) consists of 51 exons with 44 of these exons belonging to the collagenous domain (exons 3–46). Among these 44 exons, 35 exons have nucleotide numbers that are a multiple of 3 (Supplementary Table 1). When patients have truncating mutations in one of these exons, exon skipping can shift the truncation to a non-truncating mutation, that is, in-frame deletion mutations that can delay the development of ESRD in AS. This therapy of exon skipping by ASO for genetic diseases was first reported by our group in 1995 for DMD[10]. Here, we attempted exon-skipping therapy using an ASO for a truncating variant in exon 21 (84 bp) of the *COL4A5* gene.

## Results

**Split luciferase-based trimer formation of the α345(IV) proteins assay.** We have established the split nanoluciferase (NanoLuc) complementation system for examining the formation of the α345 (IV) trimer[11]. *COL4A5* plasmids for two truncating variants in exon 21, c.1350_1351delAT (p.Ile450Metfs*2) and c.1411C > T (p.Gln471*), exon 21 deletion (c.1340_1423del84bp (Δexon 21)) and the wild-type *COL4A5* were generated and examined. Although both truncating variants (p.Ile450Metfs*2 and p.Gln471*) in exon 21 failed to form trimers in cells and supernatants, the product of Δexon21 was able to form a trimer (Fig. 1).

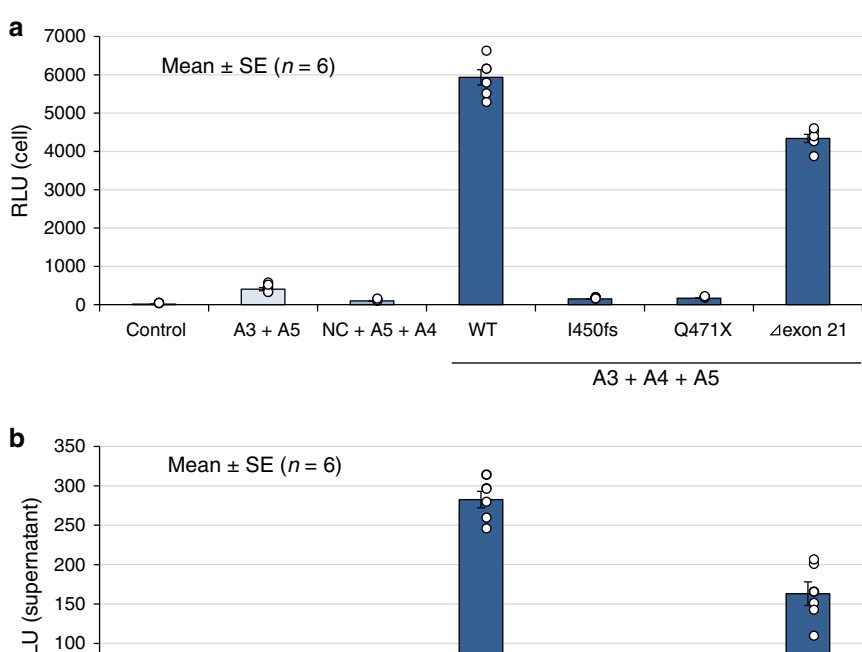

**Fig. 1 Split luciferase-based trimer formation of the α345(IV) protein assay.** Although both truncating variants, p.Ile450Metfs*2 (I450fs) and p.Gln471* (Q471X) in exon 21, failed to construct trimers, the Δexon21 variant formed trimers in both cells (**a**) and supernatant (**b**). NC: The NanoBiT® Negative Control Vector, which encodes HaloTag®-SmBiT. Error bars represent the mean ± SE. Source data are provided as a Source Data file. "*n* = 6" means that each assay was conducted 6 times.

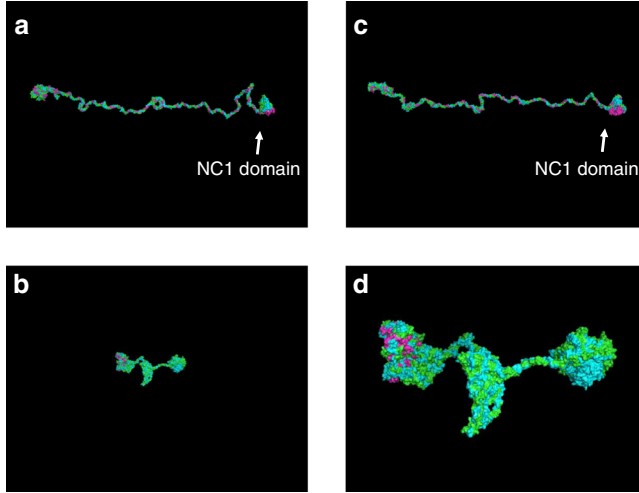

**Fig. 2 3-D structure analysis of collagen IV. a** Wild-type. **b**, **d**, p.Gln471* (**d** cropped). **c** Δexon21. Green: α3; blue: α4; and pink: α5. Formation of the trimer by p.Gln471* was followed by collapse of the whole structure. Δexon21 formed a similar structure to that of the wild-type trimer. **a–c** the same scale, **d** ×3.5 magnification to confirm the chains.

**3-D structure analysis of the α345(IV) trimer**. The 3-D structure of the α345(IV) trimer was generated using homology modeling according to our previous research with slight modifications[12]. The trimer constructed by p.Gln471* was found to aggregate, leading to a lower total potential energy. In contrast, the potential between the wild-type trimer and the trimer formed by the Δexon21 construct was similar (Fig. 2, Supplementary Fig. 1).

**In vitro exon skipping efficacy evaluation by ASO**. The most effective ASO sequence targeting exon 21 (84 bp) skipping was determined by our method[13]. We used 18-mer phosphorothioated oligonucleotides consisting of 2′-O-methyl RNA and a modified nucleic acid (2′-O, 4′-C-ethylene-bridged nucleic acid (ENA), ASO-ENA)[14]. Initially, we designed multiple series of ASOs with slightly different sequences. Each ASO-ENA was transfected individually into HEK293T cells. The ability of each ASO to induce exon skipping was examined by comparing the amount of cDNA with and without exon skipping. Finally, one ASO-ENA was determined to be the best for molecular therapy. The sequence of the ASO is: 5′-TggAguccuuuAucAccT-3′; upper letters: ENA; lower letters: 2′-O-methyl RNA. Exon skipping was also examined for patient's urine derived cultured cells (p. Gln471* in exon 21). The lowest dose (10 nM) was capable of leading to complete exon 21 skipping (Supplementary Fig. 2).

**In vivo exon skipping efficiency evaluation by ASO using a mouse model**. We have recently established a *Col4a5* mutant mouse model with c.1411C > T (p.Arg471*) in exon 21 and this mutation is equivalent to the nonsense mutation of c.1411C > T (p.Gln471*) of human *COL4A5*[15]. For this model, we initiated subcutaneous treatment with 50 mg/kg of ASO-ENA or vehicle (saline) for mutant mice (n = 6 for vehicle and n = 5 for ASO) and with vehicle for wild-type mice (n = 3). The frequency of administration was twice a week from 4 to 6 weeks of age and once a week from 7 to 20 weeks of age. Mice were sacrificed at 21 weeks of age and samples were harvested. The transcripts extracted from each mouse kidney showed clear exon 21 skipping for only mice that received ASO treatment (Supplementary Fig. 3). Immunofluorescence analysis revealed that α5(IV) was completely negative for the vehicle group but was expressed clearly on tubular basement membranes and even on the GBM, although expression is not linear but partial on GBM for the ASO group (Fig. 3; green shows α5(IV) and red shows WT1 as a podocyte marker). In addition, expression of α3(IV) and α4(IV) were observed in the ASO group with the same expression pattern of α5(IV), indicating that a collagen IV alpha345 trimer constructed after exon skipping is secreted from podocytes into the GBM (Supplementary Fig. 4). Pathological evaluation by light microscopy for hematoxylin and eosin (HE) staining revealed inflammatory cell infiltration to the interstitial area in the vehicle treated group; however, wild-type and the ASO-treated groups remained almost normal in the interstitial region (Fig. 4a–c). Masson-Trichrome staining revealed interstitial fibrosis in the vehicle treated group but not in the wild-type and ASO-treated groups (Fig. 4d–f). With periodic acid methenamine silver (PAM) staining, global sclerosis was observed in most of the glomerulus in the vehicle treated group but not in the wild-type and ASO-treated groups (Fig. 4g–i). Although ASO-treated mice showed mild irregularity of the GBM when examined by electron microscopy, they did not show severe thickening with lamellation as observed for the vehicle treated group (Fig. 4j–l). Clinical parameters showed no noticeable differences in the body weight (Fig. 5a). However, the urinary albumin creatinine ratio was remarkably suppressed in the ASO-treated group when compared with that of the vehicle treated group (Fig. 5b). At 21 weeks of age, serum BUN and creatinine levels were remarkably low in the ASO-treated group when compared with that of the vehicle treated group (Fig. 5c, d). In addition, to confirm that survival time by this therapy was prolonged, we compared the survival time between ASO or vehicle (saline) treated mutant mice (n = 8 for vehicle and ASO groups) and vehicle treated wild-type mice (n = 6). The results showed that the survival time was significantly prolonged in the ASO-treated group when compared with that of the other two groups (Fig. 5e). In conclusion, we have successfully shown the efficacy of ASO treatment with clinical–pathological findings for XLAS with truncating mutation using an animal model.

**In vivo drug delivery evaluation of ASO using a mouse model**. To confirm the uptake of ASO in kidney tissue, especially podocytes, we administrated a single dose of ASO-ENA tagged by cyanine 3 to both wild-type and mutant mice, and sacrificed the mice 24 h and 2 weeks following dosage. The results clearly showed ASO-ENA in podocytes and tubular epithelial cells of every specimen (Supplementary Fig. 5).

**Discussion**
Recently, ASO therapies have gained attention as an approach to treat various diseases[16]. Oligonucleotides are short nucleic acids that usually consist of 13–25 nucleotides and are designed to hybridize with DNA or RNA specific regions that have complementary sequences and such hybridization changes the expression level of proteins. Currently, four ASO therapies have been approved by the Food and Drug Administration (FDA): Eteplirsen for DMD, Nusinersen for SMA, Mipomersen for hypercholesterolemia and Inotersen for amyloidosis. In addition, some drugs are under clinical trials for treating chylomicronemia syndrome, Huntington's disease, and hyperlipidemias[16]. In particular, Eteplirsen is an ASO that induces exon skipping in the *DMD* gene and changes frameshift variants into in-frame variants, which leads to production of functional dystrophin. We hypothesized that applying this exon skipping therapy for XLAS would be possible because this disease also shows a strong genotype-phenotype correlation as observed for DMD. Eteplirsen has been reported to show weaker activity than expected. A

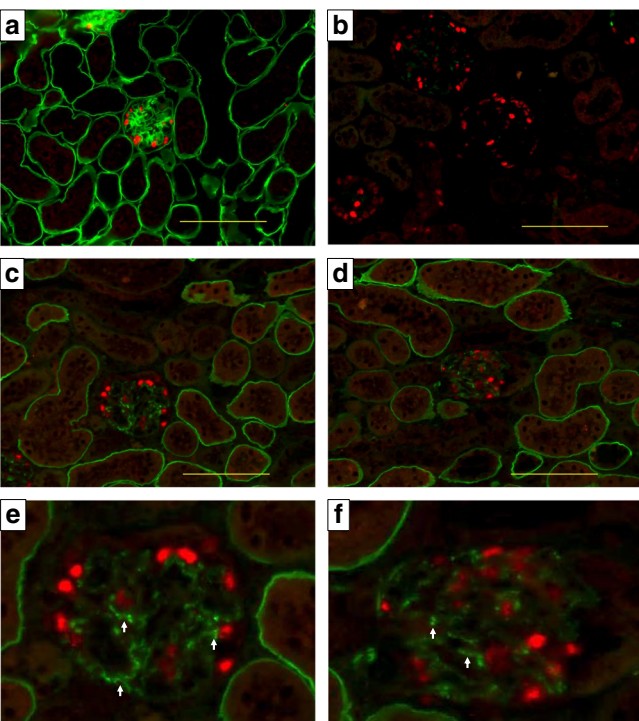

**Fig. 3 Type IV collagen α5 chain staining at 21 weeks of age.** Green: Type IV collagen α5 chain. Red: WT1 as a podocyte marker. **a** Wild-type mouse treated with the vehicle. **b** Mutant mouse treated with the vehicle. **c–f** Mutant mouse treated with the antisense oligonucleotide (ASO) (**e**, **f** cropped). Type IV collagen α5 chain was completely negative in the vehicle group (**b**), but was clearly expressed on the tubular basement membrane and even on the glomerular basement membrane (GBM); although, expression was only partial on the GBM (**e**, **f** arrows). Scale bars are 100 μm. Two different researchers from different institutes conducted the staining, and the expression pattern was identical.

reason for this weak activity may be because the majority of the drug is rapidly filtered into the urine from the glomerulus and only a small amount of the drug reaches skeletal muscle[16]. Thus, developing an ASO for the treatment of kidney diseases seems to be a very promising, straightforward approach. In this report, we showed the promising effect of ASO therapy for male XLAS. Figure 6 shows a schematic describing the possible mechanism of ASO treatment against XLAS. With truncating variants in *COL4A5*, α5(IV) is terminated at the stop codon and the NC1 domain is absent. We have shown here and previously that the NC domain is important for α345(IV) trimer formation[9,11], because formation is dependent on the association of the NC domains[3,17]. Exon-skipping therapy was found to replace the truncating variant with an in-frame deletion at the transcription level, and the resulting protein from this therapy formed the trimer. Thus, this therapy rescued progression of kidney failure in XLAS. Surprisingly, exon-skipping therapy caused clear α5(IV) expression on the tubular basement membrane and GBM. We have reported previously about 29% of the male XLAS cases show α5(IV) expression on the GBM and show much milder phenotypes of developing ESRD 13 years later when compared with cases that have negative α5(IV) expression[18]. We have also published data that splice site mutations with in-frame deletions at the transcription level show later development of ESRD[8]. These data provided impetus to develop this therapy for male XLAS cases and we have successfully proven the extraordinary effect of an exon 21 truncating variant.

This RNA targeted therapy is theoretically effective for severe male XLAS cases with truncating variants located in 35 exons in the collagenous domain with nucleotide numbers that have a multiple of three (Supplementary Table 1). Although we proved the remarkable effect of exon-skipping therapy only for exon 21 in this study, it remains unknown whether this therapeutic approach will be effective for the other 34 exons with nucleotide numbers that have multiples of three. Our previous study revealed that there is only a 9-year difference in the renal survival period between patients with splice site mutations resulting in in-frame deletions and frameshift transcript[8]. This is because in some of the exons in-frame deletions do not show milder phenotypes. Regarding exon 21, the renal survival period of the reported three male cases with estimated exon 21 skipping (c.1340-2A > G, c.1423 + 1G > A, c.1423 + 1G > T) were over 25 years, over 35 years and 52 years, respectively[19–21]. In contrast, the reported three patients with an exon 21 truncating variant (c.1376delC) reached ESRD at the mean age of 20 and there was a clear difference between these two groups[22]. In addition, we conducted an additional study to prove the substantial renal rescue effect of exon 21 skipping by a long-term mice study. The results of this study showed that the survival period was clearly prolonged in the ASO-treated group (Fig. 5e). In conclusion, these observations show that exon 21 skipping is effective.

Future alternative efforts require establishing mutant mouse models for other exons because there are another 34 exons that could be targeted by an ASO. The most promising candidate to establish a therapeutic effect evaluation system is a study using iPS cells. Currently, it is possible to generate podocytes and kidney organoids[23,24]. Using this system, we can assess the effect of each exon skipping and have initiated this effort. We are also examining whether the α345(IV) trimer is formed for all 35 exons in an effort to develop drugs for further candidate exons for this therapy using the NanoLuc system and modeling. Although our study showed no side effects to the mice, we also have to conduct good laboratory practice and a safety assessment study before initiating a clinical trial for exon 21 skipping therapy.

Targeted delivery of the drug to a particular organ is an important feature of any therapy. Targeting the kidney is advantageous because most drugs pass through the glomerulus. Our data showed tagged ENA-ASO was delivered to podocytes along with tubular epithelial cells, and the ENA-ASO remained in these cells for at least 2 weeks (Supplementary Fig. 5). This finding suggests that frequent drug administration can be avoided.

Our present study has some limitations. Firstly, unfortunately, we failed to evaluate the protein expression levels of α5(IV) in ASO-treated mice by western blotting, which may be because of the quality of the antibody. However, from other results, we concluded that exon 21 skipping showed a remarkable effect. Secondly, the exon skipping long-term side effects were not evaluated. Additionally, long-term ASO effects toward wild-type mice have not been evaluated. This treatment can cause AS with a mild phenotype. As a next step, we are planning to conduct long-term safety evaluation tests on wild-type mice and cynomolgus monkeys to assess any effects before a starting clinical trial.

In conclusion, we have developed an exon-skipping therapy that prevents progression of kidney failure in AS. These findings suggest that exon skipping represents a promising therapeutic approach for male XLAS cases with truncating mutations.

## Methods

**Split luciferase-based trimer formation of the α345(IV) proteins assay.** For examining the formation of the α345(IV) trimer, we have established a split nanoluciferase (NanoLuc) complementation system[11]. Briefly, split NanoLuc is composed of a large fragment (LgBiT) and a small fragment (SmBiT) of the NanoLuc luciferase. Luminescence is observed when the split NanoLuc-tagged proteins interact with LgBiT and SmBiT. We constructed plasmids for *COL4A3*, *COL4A4*, and *COL4A5* containing C-terminal-tagged SmBiT for *COL4A3* and

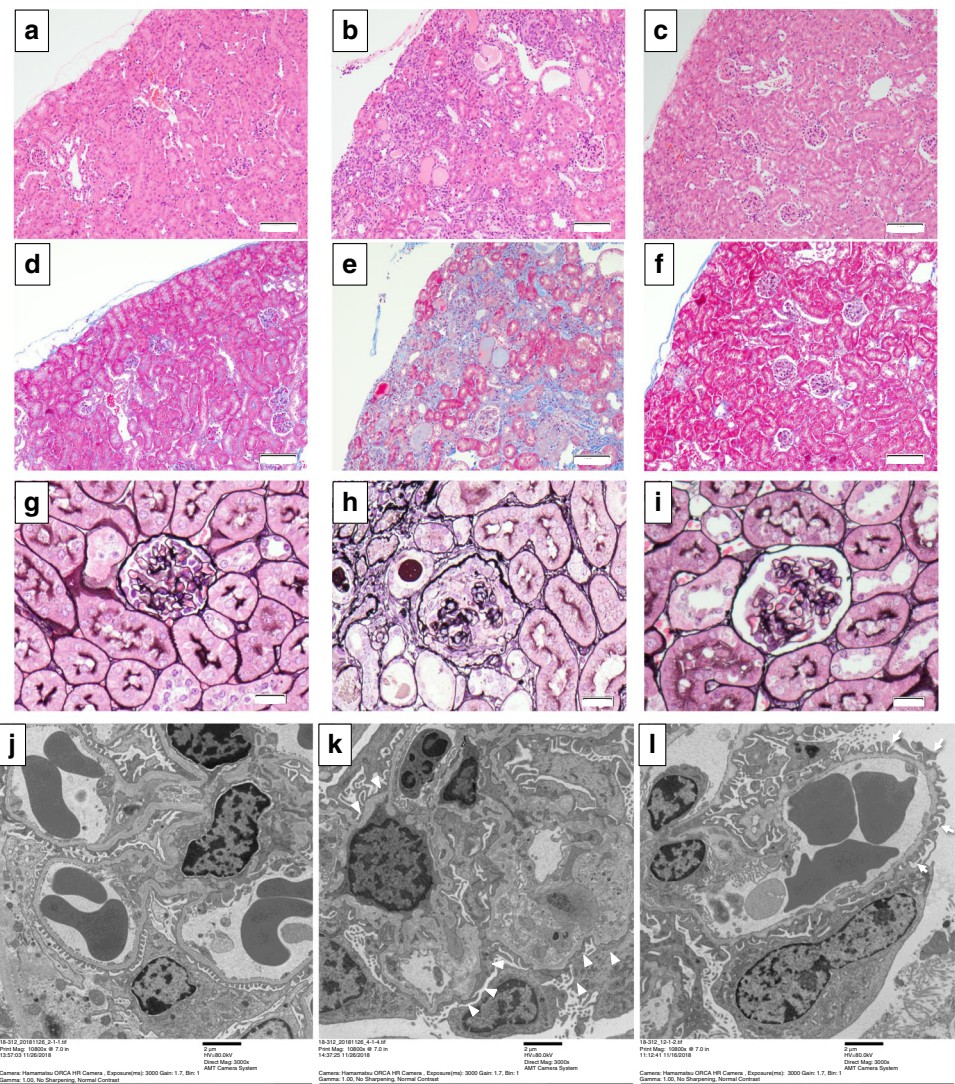

**Fig. 4 Light microscopic and electron microscopic findings at 21 weeks. a–c** Hematoxyline and eosin staining for wild-type (**a**), vehicle treated (**b**), and antisense oligonucleotide (ASO) treated (**c**) mice. Magnification, ×40. **d–f** Masson-Trichrome staining of wild-type (**d**), vehicle treated (**e**), and ASO-treated (**f**) mice. Magnification, ×40. **g–i** Periodic acid methenamine silver (PAM) staining of wild-type (**g**), vehicle treated (**h**), and ASO-treated (**i**) mice. Magnification, ×160. **j–l** Electron microscopic (EM) images of wild-type mouse treated with vehicle (**j**), mutant mouse treated with vehicle (**k**), and mutant mouse treated with the antisense oligonucleotide (**l**). With pathological evaluation by light microscopy for hematoxylin and eosin staining, inflammatory cell filtration to the interstitial area was observed in the vehicle treated group; however, wild-type (WT) and ASO-treated groups remained essentially normal in the glomerulus and interstitial region (**a–c**). Masson-Trichrome staining revealed interstitial fibrosis in the vehicle treated group but not in the WT and ASO-treated groups (**d–f**). With periodic acid methenamine silver (PAM) staining, global sclerosis was observed in most of the glomerulus in the vehicle treated group but not in the WT and ASO-treated groups (**g–i**). With EM, although ASO-treated mice showed mild irregularity of the glomerular basement membrane (GBM) (arrows), they did not show severe thickening with lamellation (L) as was observed in the vehicle treated group (**k**, arrowheads). Scale bars are 100 μm in (**a–f**), 20 μm in (**g–i**) and 2 μm in (**j–l**). Each staining was conducted once for all samples and all samples in the treatment group show identical staining patterns.

LgBiT for *COL4A5*, and those were inserted into the pBiT2.1-C [TK/SmBiT] vector and pBiT1.1-C [TK/LgBiT] vector, respectively. Following transfection of these three plasmids into HEK293T cells, luminescence was detected with high sensitivity from the heterotrimer of α345(IV) but not from the homodimer or heterodimer expressing cells or supernatant. *COL4A5* mutants were generated by mutagenesis. Primer sequences are shown in Supplementary Table 2. *COL4A5* plasmids for two truncating variants in exon 21, c.1350_1351delAT (p.Ile450Metfs*2), c.1411C > T (p.Gln471*), and exon 21 deletion (c.1340_1423del84bp (Δexon21)) were generated. *COL4A3*-SmBiT, *COL4A5*-LgBiT, and *COL4A4* plasmids were transfected into HEK293T cells. At 48 h after transfection, the Nano-Glo Live Cell Assay reagent was added and the luciferase activity in the medium and cells were measured using a SpectraMax M4 (Molecular Devices, San Jose, CA, USA). All luciferase assays were conducted in CORNING® TypeI Collagen Coated 96-Well Cell Culture Clear Bottom White plates (Nippi Incorporated). Each assay was conducted six times.

**3-D structure analysis of collagen IV**. The 3-D structure of the α345(IV) trimer was generated using homology modeling according to our previous research with slight modifications as follows[12]. The amino acid sequences of wild-type human α3 (IV)–α5(IV) were downloaded from the NCBI protein database (Accession Nos. α3, NP_000082.2; α4, NP_000083.2; α5, and NP_000486.1). The amino acid sequences of c.1411C > T (p.Gln471*) and c.1340_1423del84bp (Δexon21) mutant α5(IV) were prepared from the amino acid sequences of wild-type α5(IV). Upon a BLAST search against the Protein Data Bank (PDB), PDB IDs 3HQV, 5NB0, 5NB1, and 5NAZ were identified as templates for the triple helical region and the NC1 domains of α3(IV)–α5(IV), respectively. Since 3HQV contains only alpha carbon atoms, the alphabuild.svl script of the Molecular Operating Environment (MOE) software (Chemical Computing Group, Quebec, Canada) was used to generate the coordinates of the remaining atoms. The 3-D structures of wild-type and mutant trimers were generated using the homology modeling function in the MOE software. The resulting structures were subjected to structural optimization by

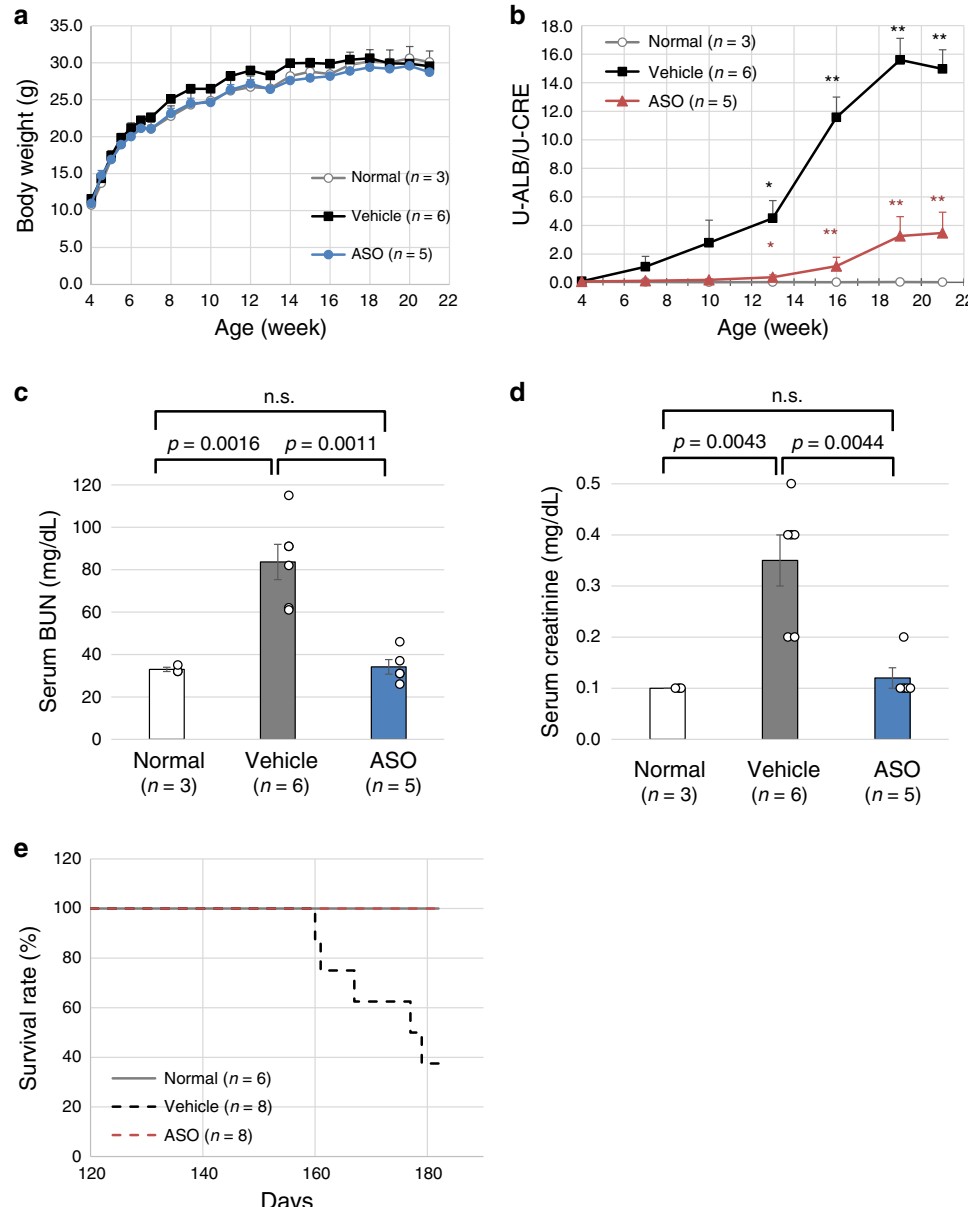

**Fig. 5 Clinical evaluation for exon skipping efficacy.** With clinical parameters, there was no difference in body weight (**a**). However, the urinary albumin creatinine ratio was remarkably suppressed in the ASO-treated group when compared with that of the vehicle treated group (**b**). * represents $p < 0.05$ and ** represents $p < 0.01$ when compared with normal mice. * in red represents $p < 0.05$ and ** in red represents $p < 0.01$ when compared with the vehicle treated mice. At 21 weeks of age, serum BUN (**c**) and creatinine levels (**d**) were remarkably low in the ASO-treated group when compared with that of the vehicle treated group. The significance of differences between two groups was assessed using the two-sided Welch's $t$ test (**b**–**d**). At the age of 177 days, 50% of the mice population had died in the vehicle treated group, whereas no mice had died in the ASO-treated group. The survival time was significantly prolonged in the ASO-treated group when compared with that of the other two groups ($P = 0.0025$ by the log-rank test) (**e**). Error bars represent the mean ± SE. Source data and exact $p$ values are provided as a Source Data file.

GROMACS version 5.1.5 (http://manual.gromacs.org/documentation/5.1-current/index.html)[12]. The AMBER99SB-ILDN force field was used for all simulations[25]. To reduce the computational cost, the protein was solvated in a generalized Born implicit solvent model instead of explicit solvent, as described in our previous studies[26,27]. To shorten the real-time required for MM/MD calculation steps of structural analysis[27], we used 94319.53 Node*Hour computational resources on the K supercomputer for structural optimization of wild-type and mutant collagen IV trimers (α3–α5). As for financial costs, the total charge of the K supercomputer was 1,370,463 yen (charge: 14.53 yen per node hour). Production runs of molecular dynamics simulations were carried out using 144 nodes on the supercomputer K (RIKEN AICS in Japan).

**In vitro exon skipping efficiency evaluation by ASO.** The most efficient ASO sequence was determined by the method reported from our group[13]. First, we

designed multiple series of ASOs with slightly different sequences. We used 18-mer phosphorothioated oligonucleotides consisting of 2′-O-methyl RNA and modified nucleic acids (2′-O, 4′-C-ethylene-bridged nucleic acid (ENA), ASO-ENA)[14]. Each ASO-ENA was transfected individually into HEK293T cells by using the Lipofectamine RNAiMAX Transfection Reagent® (Thermo Fisher Scientific), and the mRNA was analyzed by reverse transcription polymerase chain reaction (RT-PCR). Twenty-four hours later, total RNA was extracted from cells and cDNA was synthesized by using the SuperPrep® Cell lysis & RT kit for qPCR (TOYOBO). PCR was performed with a forward primer located in exon 19 and a reverse primer located in exon 23. Primer sequences are shown in Supplementary Table 2. PCR products were analyzed by LabChip™ (PerkinElmer Co., Ltd.) followed by Sanger sequencing.

**In vivo exon skipping efficiency evaluation by ASO using a mouse model.** We have recently established a *Col4a5* mutant mouse model with c.1411C > T

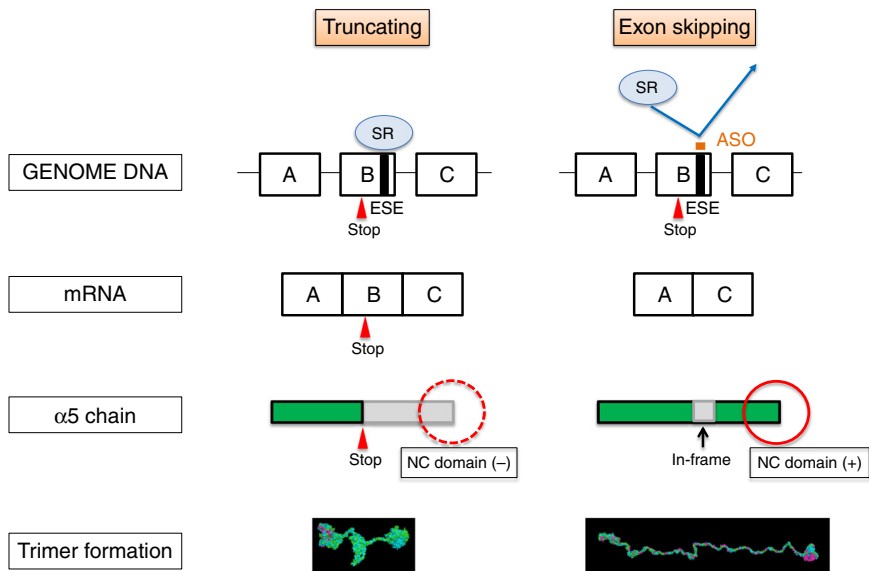

**Fig. 6 Schematic describing how the antisense oligonucleotide (ASO) treatment rescues animals from developing ESRD in XLAS.** With the truncating variants in *COL4A5*, the type collagen α5 chain will terminate at the stop codon and the NC1 domain is missing (left panel). In contrast, exon-skipping therapy will replace the truncating variant with an in-frame deletion variant at the transcript level and the NC domain is not lost, leading to the formation of the trimer for this mutation. Thus, this therapy rescued progression of kidney failure in XLAS. SR SR protein, ESE exonic splicing enhancer, NC domain non-collagenous domain.

(p.Arg471*) in exon 21 and this mutation is equivalent to the nonsense mutation of c.1411C > T (p.Gln471*) found in human *COL4A5* (developed by Axcelead Drug Discovery Partners, Kanagawa, Japan)[15]. This male mouse showed proteinuria and hematuria. The pathological finding follows human AS with diffuse GBM lamellation. These observations are consistent with the clinical and pathological features of patients with AS. These mice develop ESRD at the median age of 28 weeks. For this model, we started subcutaneous treatment with 50 mg/kg of ASO-ENA or vehicle (saline) for mutant mice ($n = 6$ for vehicle and $n = 5$ for ASO) and with vehicle for wild-type mice ($n = 3$). We started the treatment twice a week from 4 to 6 weeks of age and once a week from 7 to 20 weeks of age. We sacrificed mice at 21 weeks of age and harvested samples. All mice were kept in a regular 12 h light–12 h dark cycle under specific pathogen free conditions at 22–25 °C. We measured body weight every week and the urine albumin creatinine ratio once every 3 weeks. We also measured serum albumin, blood urea nitrogen (BUN) and creatinine levels at 21 weeks of age. For pathological evaluation, we observed light microscopic findings, electron microscopic findings and conducted immunofluorescence analysis for α3/α4/α5(IV) (129/b42/H53, Shigei Medical Research Institute, Okayama, Japan)[28], and these antibodies were used at 1:50, 1:50, and 1:100 dilutions in blocking buffer, respectively. Some were counter stained with the WT1 protein at a 100× dilution in blocking buffer (podocyte marker, Abcam ab89901, Cambridge, UK). In addition, to prove survival by exon 21 skipping was prolonged, we conducted subcutaneous injections with 50 mg/kg of ASO-ENA or vehicle (saline) for mutant mice ($n = 8$ for both vehicle and ASO) and with vehicle for wild-type mice ($n = 6$). Weekly administration was continued until 50% of vehicle treated mutant mice died.

**In vivo drug delivery evaluation of ASO using a mouse model.** For this model, we administrated a single dose of 50 mg/kg of ASO-ENA tagged with cyanine 3 at 6 weeks of age. We sacrificed mice 24 h ($n = 2$ for wild-type and $n = 3$ for mutant) or 2 weeks ($n = 2$ for both wild-type and mutant) after administration and harvested samples. All mice were kept in a regular 12 h light–12 h dark cycle under specific pathogen free conditions at 22 °C. We conducted immunofluorescence analysis for the WT1 protein at a 100× dilution in blocking buffer (podocyte marker, Abcam ab89901, Cambridge, UK) and observed uptake of ASO-ENA in kidney tissue.

**Statistical analysis.** For analysis of clinical parameters, all data are presented as means ± standard errors. The significance of differences between two groups was assessed using the Welch's *t* test. Differences with *P* values of less than 0.05 were considered statistically significant. Those were calculated by REDPOST (SAS institute Inc., Cary, NC, USA). For survival period analysis, the significance of differences between groups was assessed using the Log-rank test. Differences with *P* values of less than 0.05 were considered statistically significant.

**Reporting summary.** Further information on research design is available in the Nature Research Reporting Summary linked to this article.

## Data availability

Data that support the findings of this study are available from the corresponding author upon reasonable request. The source data underlying Figs. 1a, b, 5a–e, and Supplementary Fig. 1 as well as Supplementary Table 3 are provided as a Source Data file.

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

## Acknowledgements

We express our deepest gratitude and respect to Dr. Yoshikazu Sado for developing the finest antibody for type IV collagen α5. We thank the Edanz Group (www.edanzediting. com/ac) for editing a draft of this paper. We also thank Axcelead Drug Discovery Partners for conducting in vivo exon skipping efficiency evaluation by ASO using a mouse model.

## Author contributions

T.Y., T.H., K.L., and K.N. designed the study concept and wrote the paper. T.A., M.T., K.T., T.S., Y.O., Y.K., and M.K. designed the ASO sequence and made ASO-ENA, and conducted in vitro Split luciferase-based trimer formation of the α345(IV) proteins assay and in vivo analysis using the animal model. Y.T. and A.S. conducted 3-D structure analysis of collagen IV using supercomputer K.O., M.K., and H.K. developed Split luciferase-based trimer formation of the α345(IV) proteins assay. A.S., S.M., C.N., N.S., S.I., Y.A., N.M., R.R., M.J.Y., and Y.N. developed the gene screening system for Alport syndrome, conducted in vitro study for selecting best ASO sequence. Y.H., K.M., and S.K. follows the case and developed the urine derived cultured cells. Y.T. conducted immunofluorescent staining for alpha 3–5 as a specialist in this field. M.T. and M.M. critically reviewed the paper. All authors read and approved the final version of the paper.

## Competing interests

This study was supported by Grants-in-Aid for Scientific Research (KAKENHI) from the Ministry of Education, Culture, Sports, Science, and Technology of Japan (subject ID: 16K19642 to T.Y. and 26293203, 17H04189 to K.I., and 19K08726 to K.N.), by the Japan Agency for Medical Research and Development (AMED) (Grant nos. JP19ek0109231h0003 to K.N. and K.I., 19ek0109231s0103 to H.K., and 19ek0109231s0203 to M.T.) and Grants-in-Aid from the MEXT/JSPS Grants-in-Aid for Scientific Research (C) (Grant nos. 18K07414 to Y.T. and 19K12202 to A.S.). K.I., K.N., A.S., M.K., Y.O., K.T., and T.A. have filed a patent application on the development of antisense nucleotides for exon skipping therapy of Alport syndrome. This research used computational resources of the K supercomputer provided by the RIKEN Center for Computational Science through the HPCI System Research project (Project ID: hp180288, Y.T., A.S., and K.N.). M.M. has received consulting fees from Daiichi Sankyo Co., Ltd. K.I. has received grant support from Daiichi Sankyo Co., Ltd., consulting fees from Takeda Pharmaceutical Company and Kyowa Hakko Kirin Co., Ltd., and lecture fees from Chugai Pharmaceutical Co., Ltd., Takeda Pharmaceutical Company and Kyowa Hakko Kirin Co., Ltd. K.N. has received lecture fees from Novartis Pharmaceuticals Corporation and consulting fees from Kyowa Hakko Kirin Co., Ltd. The remaining authors declare no competing interests.

### Ethical consideration

All procedures were reviewed and approved by the Institutional Review Board of Kobe University School of Medicine (Nos. 301 and 1451). Informed consent was obtained from the patient and his parents to develop cultured urine derived cells from the patient's urine sample and conduct exon skipping therapy for the cultured cells. All animal experiments were approved by the Committee on Animal Experimentation at Daiichi Sankyo Co., Ltd. and Axcelead Drug Discovery Partners, Inc. Japan. Animals were treated in accordance with the Guide for Animal Experimentation issued from the Science Council of Japan in 2006.
