## [Peer Review File · Nature Communications]

Reviewers' Comments:

Reviewer #1:

Remarks to the Author:

In this study, Yamamura et al report a new exon skipping strategy for X-linked Alport Syndrome caused by truncating mutations in the COL4A5 gene, as a therapeutic intervention to delay disease onset. This work is based on their previous observation from a recent genotype-phenotype correlation study (Horinouchi 2018) which shows that the average age of disease onset was 9 years delayed between patients with nontruncating (20 years) and truncating (29 years) splice site mutations. In this study, they investigated exon-skipping ASO for a truncating variant in exon 21 (84bp) in both minigene construct and in patient's urine derived cultured cells. Furthermore, the efficacy of this ASO was validated in a mouse model carrying the same mutation. The in vivo data shows the significant rescue of this ASO on kidney pathology and the related metabolic in urine and serum in the treated mice. The manuscript is concise, experiments are well constructed and methods are sound.

Here are some comments for authors:

1. The predicted 3-D structure of the COLIV α 3/4/5 with exon 21 skipping is very close to wild-type. The rescue of ASO in kidney pathology and function is also very promising to near wild-type level. Based on these striking rescue, one would presume that the treated patients with induced-in-frame deletion may have much milder or near normal phenotype. In fact, however, patients carrying splice-site mutations-with in-frame transcripts still show the moderate disease phenotypes, with an average age of onset at 29 years old-that still seems severe when compared with patients carrying missense mutations whose average age at onset is 37 years-, I wonder if the author may add more discussion on this?

2. Other minor comments:

a. Page 5, line 96: 1995 instead of 1985.

b. Page 9, line 161: APO>ASO.

c. Page 10, line 174: Figure 6, not 7.

d. Figure legends- Figure 3: please indicate what the staining in red is?

e. Please add n=?/group in Figure 5.

Reviewer #2:

Remarks to the Author:

For Authors

This is an important paper showing the benefit of forced exon skipping in a specially designed mouse model of Alport syndrome that was previously published. The major claim of the paper is very well supported: treatment of the mice with an oligonucleotide that promotes skipping of an 84 bp exon with an engineered nonsense codon results in production and secretion of collagen IV α 5, presumably within collagen IV α 3/4/5 heterotrimers, though this is not demonstrated. Evidence is presented showing that collagen IV α 5 is present in the tubular basement membranes of treated Alport mice, and less so in the glomerular basement membranes where this collagen is known to be most important to maintain kidney function. Despite the low glomerular expression, there is a dramatic positive impact on kidney histopathology at the light and electron microscopic levels. Also, the levels of urinary albumin, BUN, and serum creatinine are significantly reduced in the oligo-treated mice compared to those treated with vehicle. Because many patients with Alport syndrome have nonsense mutations in exons whose base pair sizes are multiples of 3, this treatment approach could be very beneficial for them, especially because the benefit of treatment was significant though the level of the collagen IV restored to the GBM might be much less than in normal.

Although the overall result will be of great interest to the nephrology community and to other researchers who study diseases caused by nonsense mutations, the paper provides only a

superficial analysis of the mice and the impact of the treatment. There are a number of obvious questions that should have been answered by the authors for this high impact journal.

Major issues

1. Fig. 3 is very important. The authors should provide higher power images of the glomeruli of treated Alport mice so readers can better see how much COL4 alpha5 is in the GBM and to what extent. In addition, it is important to show that the partial protein that is made after exon skipping is secreted as part of a collagen IV alpha345 trimer; this can easily be done by staining for alpha4 and/or alpha3, which should also be absent from the vehicle-treated basement membranes. Finally, a western blot quantification of Collagen IV alpha5 (or alpha3 or 4) in treated Alport vs. untreated WT kidney is important because the dramatic improvement in phenotype needs to be correlated to the degree of protein secretion rescue. When considering treating patients, knowing (for example) that 10% restoration of protein in the mouse is effective would be very useful.
2. In Supp Fig. 4, some of the arrows are pointing to mesangial cells or endothelial cells rather than to podocytes. It would be more useful to stain the section for a podocyte marker such as synaptopodin to better identify oligo-positive podocytes. Can this be quantified? After how many injections was the mouse sacrificed? Might podocytes in the mutant be targeted more than in WT due to the glomerular disease?
3. It would be very useful to know the effect of the ASO treatment on a wild-type mouse. Would it cause an Alport syndrome-like kidney disease? Did the authors attempt this?

Minor issues

1. This manuscript refers to the mouse Alport mutation as p.Gln471* in exon 21, but the original paper reporting this mouse calls it R471X (or p.Arg471*). Which is correct?
2. In Supp Fig 3, the lanes from Col4a5 mutant mice should be indicated.

Reviewer #3:

Remarks to the Author:

The paper presents the development of an exon-skipping therapy using an antisense-oligonucleotide for male X-linked Alport syndrome. With respect to aspects of molecular modelling, which are within the expertise of the reviewer, the 3D structure of the alpha345(IV) trimer for the different variants herein addressed has been analysed by means of homology modelling. The online methods are clear in this aspect and results are convincing.

The reviewer does not have main issue to arise, apart some minor remarks:

- Information on the computational cost are missing.
- Discussions on the effect of using an implicit solvent model instead of an explicit solvent are missing. References supporting that results are not highly affected by this choice would be needed.
- The clarity of the label of Fig. 2 might be improved by better specifying the difference between plot in b-1 and b-2 (the reviewer understood what Authors mean but it is not obvious).
- The mutant version is referred to Q471X in Supplementary Figure 1b. This is never mentioned in the main text and only introduced in the labels of Fig. 1. The reviewer would change it to avoid confusion.

Our point-by-point responses to the comments raised by the three reviewers are described on the following pages.

Responses to the comments raised by Reviewer #1:

Reviewer #1 (Remarks to the Author):

In this study, Yamamura et al report a new exon skipping strategy for X-linked Alport Syndrome caused by truncating mutations in the COL4A5 gene, as a therapeutic intervention to delay disease onset. This work is based on their previous observation from a recent genotype-phenotype correlation study (Horinouchi 2018) which shows that the average age of disease onset was 9 years delayed between patients with nontruncating (20 years) and truncating (29 years) splice site mutations. In this study, they investigated exon-skipping ASO for a truncating variant in exon 21 (84bp) in both minigene construct and in patient's urine derived cultured cells. Furthermore, the efficacy of this ASO was validated in a mouse model carrying the same mutation. The in vivo data shows the significant rescue of this ASO on kidney pathology and the related metabolic in urine and serum in the treated mice. The manuscript is concise, experiments are well constructed and methods are sound. Here are some comments for authors:

We appreciate the positive comments and feedback from Reviewer #1. We have responded to each point below.

1. The predicted 3-D structure of the COLIVa3/4/5 with exon 21 skipping is very close to wild-type. The rescue of ASO in kidney pathology and function is also very promising to near wild-type level. Based on these striking rescue, one would presume that the treated patients with induced-in-frame deletion may have much milder or near normal phenotype. In fact, however, patients carrying splice-site mutations-with in-frame transcripts still show the moderate disease phenotypes, with an average age of onset at 29 years old-that still seems severe when compared with patients carrying missense mutations whose average age at onset is 37 years-, I wonder if the author may add more discussion on this?

Thank you for your positive evaluation and comment. We also understand that this point is very important for future possible use of the exon skipping approach for XLAS treatment. Although our team conducted genotype-phenotype correlation analysis for patients with splicing variants and revealed an 8 year difference in the renal survival period between patients with truncating and non-truncating transcript in a previous study, we think this severity change may be dependent on each exon; that is, in some of the exons an in-frame deletion did not lead to milder phenotypes for patients, such as in exon 9 (unpublished data). Regarding exon 21, the rescue effect of exon skipping is thought to be very large because renal phenotypes of some reported cases with estimated exon 21 skipping are very mild. The renal survival period of the three reported male cases with estimated exon 21 skipping (c.1340-2A>G, c.1423+1G>A, c.1423+1G>T) are over 25 years, over 35 years and 52 years, respectively (PMID 9195222, 9848783, 30773290). In contrast, the reported three patients with an exon 21 truncating variant (c.1376delC) reached ESRD at a mean age of 20 (PMID 8940267).

Thus, there is a big difference between these groups. In addition, we conducted a long-term mice study to prove the substantial renal rescue effect of exon 21 skipping. The results of this study showed that the survival period was prolonged considerably in the ASO treated group (Fig. 6e). Based on these new observations, we conclude that exon 21 skipping is effective. We have added experimental details, observations and discussion in the Methods, Results and Discussion sections, respectively.

2. Other minor comments:

a. Page 5, line 96: 1995 instead of 1985.

b. Page 9, line 161: APO>ASO.

c. Page 10, line 174: Figure 6, not 7.

Thank you for the comments. We have corrected these errors.

d. Figure legends- Figure 3: please indicate what the staining in red is?

Thank you for the comment. Red shows the WT1 protein and we have added details to the legend and text.

e. Please add n=?/group in Figure 5.

Thank you for the comment. We have added the number of each group to Figure 5.

Responses to the comments raised by Reviewer #2:

Reviewer #2 (Remarks to the Author):

For Authors

This is an important paper showing the benefit of forced exon skipping in a specially designed mouse model of Alport syndrome that was previously published. The major claim of the paper is very well supported: treatment of the mice with an oligonucleotide that promotes skipping of an 84 bp exon with an engineered nonsense codon results in production and secretion of collagen IV alpha5, presumably within collagen IV alpha3/4/5 heterotrimers, though this is not demonstrated. Evidence is presented showing that collagen IV alpha5 is present in the tubular basement membranes of treated Alport mice, and less so in the glomerular basement membranes where this collagen is known to be most important to maintain kidney function. Despite the low glomerular expression, there is a dramatic positive impact on kidney histopathology at the light and electron microscopic levels. Also, the levels of urinary albumin, BUN, and serum creatinine are significantly reduced in the oligo-treated mice compared to those treated with vehicle. Because many patients with Alport syndrome have nonsense mutations in exons whose base pair sizes are multiples of 3, this treatment approach could be very beneficial for them, especially because the benefit of treatment was significant though the level of the collagen IV restored to the GBM might be much less than in normal.

Although the overall result will be of great interest to the nephrology community and to other researchers who study diseases caused by nonsense mutations, the paper provides only a superficial analysis of the mice and the impact of the treatment. There are a number of obvious questions that should have been answered by the authors for this high impact journal.

We appreciate the positive comments and feedback from Reviewer #2. We have responded to each point below.

Major issues

1. Fig. 3 is very important. The authors should provide higher power images of the glomeruli of treated Alport mice so readers can better see how much COL4 alpha5 is in the GBM and to what extent. In addition, it is important to show that the partial protein that is made after exon skipping is secreted as part of a collagen IV alpha345 trimer; this can easily be done by staining for alpha4 and/or alpha3, which should also be absent from the vehicle-treated basement membranes. Finally, a western blot quantification of Collagen IV alpha5 (or alpha3 or 4) in treated Alport vs. untreated WT kidney is important because the dramatic improvement in phenotype needs to be correlated to the degree of protein secretion rescue. When considering treating patients, knowing (for example) that 10% restoration of protein in the mouse is effective would be very useful.

Thank you for these comments. We have added higher power images (Figure 3, panels c-3 and c-4) to show more detail of $\alpha 5$ expression in glomeruli. In addition, we have conducted staining for $\alpha 3$ and $\alpha 4$ chains (Figure 4). As a result, the expression of both $\alpha 3$ and $\alpha 4$ chains were partially positive in ASO treated mice. We think antibodies for $\alpha 3$ and $\alpha 4$ chains are not good enough but we do observe differences in protein expression between treated and untreated groups. We have added text describing these results as recommended by Reviewer #2. Unfortunately, we made several attempts to blot $\alpha 5$ but failed even in wild-type mice. This current failure to successfully blot $\alpha 5$ may be because of the quality of the antibody. We hope the reviewer will accept this. We believe that other additional data compensate this missing result to prove the effect of this exon skipping therapy for this point.

2. In Supp Fig. 4, some of the arrows are pointing to mesangial cells or endothelial cells rather than to podocytes. It would be more useful to stain the section for a podocyte marker such as synaptopodin to better identify oligo-positive podocytes. Can this be quantified? After how many injections was the mouse sacrificed? Might podocytes in the mutant be targeted more than in WT due to the glomerular disease?

Thank you for this very important point. To confirm the uptake of Cy3 tagged-ASO in podocytes in mutant mice, we injected 50 mg/kg s.c. only once and sacrificed mice after 24 h and 2 weeks. We used a WT1 antibody as a podocyte marker. The results showed clear uptake of ASO by podocytes and tubular epithelial cells in every specimen even after 2 weeks, and we have changed Supplementary Figure 4 to reflect these results. We also tried to quantify the oligo positive podocytes; however, it was difficult to ascertain whether the uptake of ASO occurred in podocytes or other surrounding cells in some of the glomerulus. We also compared this uptake between wild-type and mutant cells. No differences were observed because uptake of the drug was seen for most of the podocytes. In the originally submitted version, we only put these data in the discussion as a reference. However, in this revised version, we placed text in the methods and the results in applicable sections.

3. It would be very useful to know the effect of the ASO treatment on a wild-type mouse. Would it cause an Alport syndrome-like kidney disease? Did the authors attempt this?

Thank you for the comment. We injected ASO once at 50 mg/kg s.c into wild type mice (6 weeks) and checked the urine albumin after 1 and 2 weeks. The results showed that urine albumin was negative. However, these are preliminary data and we did not conduct long-term observations for this report. Nonetheless, as a next step, we are planning to conduct long-term safety evaluation tests using wild-type mice and cynomolgus monkeys. We have added comments regarding these points to the Discussion.

Minor issues

1. This manuscript refers to the mouse Alport mutation as p.Gln471* in exon 21, but the original paper reporting this mouse calls it R471X (or p.Arg471*). Which is correct?

Thank you for raising this error. The amino acid is different at p.471 between mice and human. p.Arg471* (c.1411C>T) is correct for mouse *Col4a5* gene and p.Gln471* is correct for human *COL4A5* gene. We corrected these points.

2. In Supp Fig 3, the lanes from Col4a5 mutant mice should be indicated.

Thank you for this point. We have corrected this.

Responses to the comments raised by reviewer #3:

Reviewer #3 (Remarks to the Author):

The paper presents the development of an exon-skipping therapy using an antisense-oligonucleotide for male X-linked Alport syndrome. With respect to aspects of molecular modelling, which are within the expertise of the reviewer, the 3D structure of the alpha345(IV) trimer for the different variants herein addressed has been analysed by means of homology modelling. The online methods are clear in this aspect and results are convincing.

The reviewer does not have main issue to arise, apart some minor remarks:

We appreciate the positive comments and feedback from Reviewer #3. We have responded to each point below.

- Information on the computational cost are missing.

We used 94319.53 Node*Hour computational resources for structural optimization of wild-type and mutant collagen IV trimers ($\alpha 3$, $\alpha 4$ and $\alpha 5$). As for financial costs, the total charge of the K supercomputer was 1,370,463 yen (charge: 14.53 yen per node hour). We put the text about this information.

Reference 27 has been added to explain the computational cost of the K computer.

- Discussions on the effect of using an implicit solvent model instead of an explicit solvent are missing. References supporting that results are not highly affected by this choice would be needed.

A previous manuscript using the Born implicit solvent model was added as reference 26. This model is described in the “Supplementary Material” of the reference 26

- The clarity of the label of Fig. 2 might be improved by better specifying the difference between plot in b-1 and b-2 (the reviewer understood what Authors mean but it is not obvious).

To clarify the difference between the plot in b-1 and b-2, we have added an explanation to the Figure legend: “a, b-1 and c, the same scale; b-2, $\times 3.5$ magnification to confirm the chains.”

- The mutant version is referred to Q471X in Supplementary Figure 1b. This is never mentioned in the main text and only introduced in the labels of Fig. 1. The reviewer would change it to avoid confusion.

Thanks for this important point.

The nonsense mutation Q471X (Gln471*) of the human *COL4A5* gene is equivalent to the nonsense mutation of R471X (Arg471*) of the mouse *Col4a5* gene. Therefore, we have evaluated Q471X in silico and in vitro and R471X in vivo. We have added an explanation in the main body of the manuscript.

Reviewers' Comments:

Reviewer #1:

Remarks to the Author:

The author has thoroughly addressed all the queries I had in the last version, and provided more details in the revised version. I am satisfied with the revision without any further comments.

Thank you.

Reviewer #2:

Remarks to the Author:

The authors have addressed my major comments by showing higher power images in Fig. 3, which are adequate, and attempting to show basement membrane staining for alpha3 and alpha 4 type IV collagen chains, which should be present in basement membranes of the ASO treated mutant if the alpha5 chain is expressed and assembled and secreted when the mutated exon is skipped. They used three Shigei antibodies, which have been shown to stain collagen IV chains in basement membranes of human and mouse kidneys. The authors must realize that the panels marked "Normal" in Fig. 4 should all look the same because alpha3, 4, and 5 are in the same trimers (except Bowman's capsule around the glomerulus). But these 3 panels do not all look similar to each other, so I conclude that the positive control staining did not work for alpha3 and 4. This means the results in the other panels cannot be taken as valid. Also, the mostly mesangial staining for alpha3 and 4 in the glomerulus is not consistent with the known localization of these chains. Finally, I'll note that the alpha5 Normal panel in Fig. 4 was taken from the panel in Fig. 3a without indicating this fact in the legend.

I cannot support publication of manuscript with technical flaws. Whether alpha3 and alpha4 are co-secreted with the mutant alpha5 is an important question that is still not answered. I hope the authors can find a way to perform the necessary antibody staining in their mice.

Reviewer #3:

Remarks to the Author:

The Authors have accomplished all my previous remarks. Therefore, in reviewer's opinion, the paper can be published in its present form.

Our point-by-point responses to the reviewer's comments are described in the following page.

Reply to Reviewers#2' comments:

Reviewer #2 (Remarks to the Author):

The authors have addressed my major comments by showing higher power images in Fig. 3, which are adequate, and attempting to show basement membrane staining for alpha3 and alpha 4 type IV collagen chains, which should be present in basement membranes of the ASO treated mutant if the alpha5 chain is expressed and assembled and secreted when the mutated exon is skipped. They used three Shigei antibodies, which have been shown to stain collagen IV chains in basement membranes of human and mouse kidneys. The authors must realize that the panels marked "Normal" in Fig. 4 should all look the same because alpha3, 4, and 5 are in the same trimers (except Bowman's capsule around the glomerulus). But these 3 panels do not all look similar to each other, so I conclude that the positive control staining did not work for alpha3 and 4. This means the results in the other panels cannot be taken as valid. Also, the mostly mesangial staining for alpha3 and 4 in the glomerulus is not consistent with the known localization of these chains. Finally, I'll note that the alpha5 Normal panel in Fig. 4 was taken from the panel in Fig. 3a without indicating this fact in the legend.

I cannot support publication of manuscript with technical flaws. Whether alpha3 and alpha4 are co-secreted with the mutant alpha5 is an important question that is still not answered. I hope the authors can find a way to perform the necessary antibody staining in their mice.

Thank you for the comments. As pointed out, positive control staining of alpha3 and 4 was inadequate. Therefore, we invited Dr. Yasuko Tomono from the Shigei Medical Research Institute as a collaborator and asked her to stain our samples by using appropriate antibodies. She changed antibodies for alpha3 from H31 to 129 and for alpha4 from H43 to b42. As a result, these antibodies worked superbly and support our results. As we expected and as commented by reviewer #2, alphas 3–5 are stained equally in the wild-type glomerulus (Supplementary Fig. 4a–f) and mutant mice treated with saline showed no expression of these three chains (Supplementary Fig. 4g–l). In contrast, mutant mice treated with ASO showed similar levels of partial expression of these three chains on the GBM and tubular basement membrane (Supplementary Fig. 4m–r). We changed the figures by providing a new Supplementary Figure 4 and have added Dr. Tomono as a coauthor. We also put texts to Result and Method regarding this point. We believe these changes have improved our study and clearly show the treatment effect of the exon-skipping therapy. We really appreciate the comments raised by reviewer 2.

Reviewers' Comments:

Reviewer #2:

Remarks to the Author:

The authors have achieved great staining for collagen IV chains. The results support their conclusions.